# Effect of Supplementing Hydrolysable Tannins to a Grower–Finisher Diet Containing Divergent PUFA Levels on Growth Performance, Boar Taint Levels in Back Fat and Intestinal Microbiota of Entire Males

**DOI:** 10.3390/ani9121063

**Published:** 2019-12-02

**Authors:** Marco Tretola, Federica Maghin, Paolo Silacci, Silvia Ampuero, Giuseppe Bee

**Affiliations:** 1Agroscope, Institute for Livestock Sciences, La Tioleyre 4, 1725 Posieux, Switzerland; marco.tretola@agroscope.admin.ch (M.T.); paolo.silacci@agroscope.admin.ch (P.S.); silvia.ampuero@agroscope.admin.ch (S.A.); 2Department of Health, Animal Science and Food Safety, VESPA, University of Milan, 20133 Milano, Italy; federica.maghin@gmail.com

**Keywords:** hydrolysable tannins, PUFA, gut microbiota, gut health, boar taint

## Abstract

**Simple Summary:**

The European pig production industry needs to adapt to societal concerns regarding animal welfare. One of these concerns is the current castration practice of male piglets, which is primarily performed to avoid boar taint. Boar taint is an unpleasant and unwanted odor and flavor that can be found in the pork of entire male pigs. The compounds skatole, indole, and androstenone are the primary causes of boar taint, which can occur during cooking when the compounds evaporate. Recently, it has been hypothesized that with increasing levels of polyunsaturated fatty acids in the adipose tissue, the concentration of boar taint compounds decreases. Furthermore, we have shown that hydrolysable tannins included in pig diets tend to reduce the level of boar taint compounds in the adipose tissue. Thus, the objective of this study was to determine whether cumulative effects on the level of boar taint can be expected when both polyunsaturated fatty acids and hydrolysable tannins rich compounds are included in the diet. In addition, special emphasis was put on their effects on the gut microbiota composition. Investigating the gut bacterial population, we observed that polyunsaturated fatty acids had no effects on gut microbiota, while hydrolysable tannins affected both the quantity and quality of bacteria in a way that could explain the observed boar taint levels.

**Abstract:**

A retrospective data analysis suggested that the levels of boar taint compounds depend on the polyunsaturated fatty acid (PUFA) level of the adipose tissue (AT) being significantly greater in the unsaturated AT. In addition, we recently reported that hydrolysable tannins (HTs) offered to entire males (EMs) reduce skatole and, to a greater extent, indole levels in the AT. Thus, the objective of the study was to determine the impact of HTs and a high dietary level of PUFA on growth performance and board taint compounds in EMs. In addition, the interaction between PUFA and HTs on gut microbiota and its link to intestinal skatole and indole production was investigated. At 25 kg BW, 44 EM originating from 11 litters were randomly assigned within litter to four dietary treatments. Two basal grower (25–60 kg BW) and finisher (60–105 kg BW) diets containing either 2% soy oil (H = high PUFA level) or 2% tallow (L = low PUFA level) were formulated. The H and L diets were either supplemented (H+/L+) or not (H−/L−) with 3% chestnut extract containing 50% HTs. The pigs had ad libitum access to the diets and were slaughtered at 170 days of age. The microbiota composition was investigated through the 16S rRNA gene sequences obtained by next-generation sequencing (Illumia MiSeq platform, San Diego, CA, USA) and analyzed with a specific packages in R, version 3.5.0. Regardless of the PUFA content, the EMs fed the H+ diets were 2% (*p* < 0.01) less feed efficient overall. This was due to the slower (*p* = 0.01) growth in the finisher period despite similar feed intake. Carcass characteristics were not affected by the diets. Regardless of HT feeding, the PUFA level in the AT of the H pigs was 10% greater (*p* = 0.05) than in the L pigs. The indole level tended (*p* = 0.08) to be 50% lower in the H+ group. Surprisingly, the pigs that were fed diet H− had greater skatole levels than those fed diet L−, with intermediate skatole levels in the H+ and L+. Independent of the PUFA level, the HTs decreased bacteria abundance and qualitatively affected the microbiota composition. In conclusion, these data do not confirm that boar taint compound levels were related to PUFA levels in the AT. However, HTs can be considered to be a promising alternative to conventional antibacterial additives, with no detrimental effects on pig gut health and with appealing properties for reducing the synthesis of the main components of boar taint.

## 1. Introduction

The surgical castration of male pigs without anaesthesia and analgesia is a major welfare concern in Europe, and the search for viable alternatives is an ongoing process. Raising entire males (EMs) is one of the envisioned alternatives, but despite positive aspects regarding growth efficiency and carcass characteristics, major challenges arise from the incidence of boar taint [1]. Boar taint, depicted as a urinary and fecal odor and flavor, originates primarily from the testicular steroid androstenone (A), and from skatole (S) and indole (I), metabolites derived from the bacterial catabolism of tryptophan in the hindgut. These compounds are lipophilic, accumulate in the lipid fraction and, consequently, have a negative effect on the odor and flavor of pork from EMs and ultimately consumer acceptance [2].

Wesoly and Weiler [3] reviewed the effects of various feed ingredients (e.g., chicory, raw potato starch, and sugar beet pulp) on microbial metabolism, and the results indicated a change from a primarily proteolytic to a primarily saccharolytic metabolism in the hindgut. Consequently, less S and I are produced and ultimately incorporated into pork from EMs. The results of recent studies have suggested that bioactive compounds, such as hydrolysable tannins (HTs), included in the diets of finisher pigs also have the potential to reduce bacteria-mediated S and I production in the colon, resulting in lower tissue levels of the two boar taint compounds in the adipose tissue [4,5]. In line with these findings, Bilić-Šobot et al. [6] reported the lower apoptosis of intestinal epithelial cells, the main source of L-tryptophan for microbial indolic compound production, in the EMs fed a diet supplemented with 3% HTs.

Until recently, no evidence had suggested that A tissue concentration was affected by nutrition [7]. However, Jen and Squires [8] found that the EMs fed with activated carbon for 28 days in the finishing period had lower plasma as well as tissue A levels compared to a control group. The authors hypothesized that A, like oestadriol, undergoes enterohepatic circulation [9], and activated carbon could act as an absorbent, thereby diminishing A tissue levels. Recently, Bee et al. [5] observed that A concentration linearly decreased with increasing dietary HT intake. However, the HT supplemented diet in these studies was fed only in the finisher and not in the grower period. The authors hypothesized that if HT interferes with the development of androgen secretion and is offered to the pig in early life, it may be possible to decrease A levels before pigs reach puberty. Furthermore, Moerlein and Tholen [10] reviewed data from published studies and observed that EMs with low A, S, and I levels had more unsaturated adipose tissue. In pigs, the fatty acid composition of the lipids can be easily manipulated by nutrition, and the aforementioned findings give rise to the interesting possibility of indirectly controlling (via the lipid metabolism) the A and/or S levels in the adipose tissue of EMs. Consequently, we hypothesized that the inclusion of HTs and a fat source rich in PUFA in the grower and finisher diet could diminish boar taint compounds in the adipose tissue and in the intramuscular fat. Thus, the aims of this study were to investigate the effects of dietary HT and high and low PUFA supplementation on A, S, and I tissue levels in Ems, and the possible regulation of hepatic clearance of A and S. In addition, the quantitative and qualitative modifications of gut microbiota were investigated.

## 2. Materials and Methods

### 2.1. Animals and Diets

The Swiss cantonal Committee for Animal Care and Use approved all the procedures involving animals (27,428).

Forty-four Swiss Large White EMs selected from the Agroscope sow herd and originating from 11 litters (on average 26 ± 1.5 days old) with an average weight of 26.0 ± 4.95 kg (average ± standard deviation) were selected and randomly assigned within litter to four dietary treatments: a high amount of PUFA without chestnut extract containing HTs (H−), a high amount of PUFA with 3% chestnut extract containing HTs (H+), a low amount of PUFA without chestnut extract containing HTs (L−), and a low amount of PUFA with 3% chestnut extract containing HTs (L+) (Table 1).

The chestnut extract (*Castanea sativa*) (Silvateam, San Michele Mondovì Italy) contained approximately 50% HTs [5]. To increase the PUFA content in the diet soy oil. All the diets were formulated to be isocaloric and isonitrogenous and to meet nutrient requirements according to the Swiss feeding recommendations for pigs (Agroscope 2019). The experimental diets were offered ad libitum in a pelleted form on average for 98 d. All the pigs were reared in group pens equipped with automatic feeders and an individual pig recognition system (Schauer Maschinenfabrik GmbH. & Co KG, Prambachkirchen, Austria), which allowed the authors to monitor individual daily feed intake [11]. In addition, the individual feeding records were compiled to assess feeder feeding behavior traits, such as total and daily visits to the feeder, feed intake per visit and per minute, and time spent at the feeder in total, per day and per visit. When the average BW of all 44 pigs was 60 kg, the pigs were switched from the grower to the finisher diets.

### 2.2. Slaughter Procedure, Carcass Measurements, Tissue Sampling, and Dual-Energy X-ray Absorptiometry (DXA) Measurements

The pigs were slaughtered at 171 ± 2.8 d of age at the research station abattoir after being fasted for approximately 15 h. A detailed description of the slaughter and sampling methods was previously presented by Bee et al. [5]. Briefly, 30 min after exsanguinations, the weights of the hot carcasses, livers, kidneys, testicles, and salivary (mandibular), bulbo-urethral, and parotids glands were assessed. In addition, the content of the caecum was collected and immediately frozen in liquid nitrogen and stored at −80 °C until analysis. Subsequently, the hot carcass weight and muscle pH at 45 min post-mortem was determined before they were chilled at 2 °C for 24 h. One day post mortem, the left cold carcass weight was determined and subsequently dissected into the major primal cuts (loin, ham, shoulder, and belly). The carcass yield, expressed as the proportion of the hot carcass weight over the BW at slaughter, was calculated. The lean and back fat percentages were calculated as the proportion of the defatted primal cuts over the cold carcass weight and the sum of the fat cuts from the primal cuts over the cold carcass weight, respectively.

The carcass composition was assessed by DXA using the GE Lunar iDXA (Bucks, UK) with the GE Encore software package (v. 17 SP4), both provided by the same company (GE Healthcare, Glattbrugg, Switzerland). Prior to carcass scanning, a calibration procedure using the manufacturer’s recommendations was performed. With the iDXA device, the bone mineral, fat, and lean masses were determined.

### 2.3. Meat Quality Traits

After the carcass dissection was completed, and following the protocol described by Pardo et al. [12], the longissimus thoracis muscle (LT) at the 10th rib level was removed for the determination of ultimate pH, meat color (L*, a, b* values), water holding capacity as a percentage drip after 24 h, and cooking and thaw loss and shear force. Briefly, five 2.5 cm chops weighing on average 80 g each were used. One chop was freed of visible fat, vacuum-packaged and stored at −20° until the chemical analysis. Color measurements were performed by using the Chroma Meter CR-300 with a D65 light source (Minolta, Dietikon, Switzerland) after 20 min blooming using 2 chops. The same chops were then used for assessing 24 h drip loss using the bag method. The remaining two slices were weighed, vacuum-packaged, and stored at −20 °C for the determination of thaw and cooking loss, and shear force.

### 2.4. Feed and Meat Analysis

The feed and LT samples, after being milled with a 1 mm sieve and freeze-dried, respectively, were analyzed for dry matter and ash content by, respectively, heating at 105 °C for 3 h and incinerating at 550 °C until reaching a constant mass. Using the Kjeldahl procedure (Leco FP-2000 analyser, Leco, Mönchengladbach, Germany), the N content of the feed was analyzed, and the CP content was expressed as 6.25 × N. The content of crude fiber was analyzed after successive digestion with H_2_SO_4_ and KOH, washed with acetone, dried at 130 °C and then ashed (EN 71/393, ISO 6865:2000, VDLUFA 6.1.4). The fatty acid profile of the diets and adipose tissue was obtained by gas chromatography (GC) with in situ methylation transesterification, as previously described in detail [13]. Briefly, 250 mg of the freeze-dried sample was placed in a polytetrafluoroethylene tube with 1 mL of internal standard (1 mg/mL C19:0 in toluene) and 3 mL of 5% HCl in methanol. After 3 h at 70 °C under gentle mixing, the reaction mix was neutralized at room temperature with 5 mL of 6% K_2_CO_3_. After the addition of 3 mL pentane and centrifugation (5 min at 2500 rpm) the organic phase was dried with ~3 g of dehydrated Na_2_SO_3_ and ~0.2 g of activated charcoal. After mixing well, the tube was allowed to stand still for 1 h, then it was centrifuged (5 min at 2500 rpm), and the organic phase was dried out at 40 °C under a constant flux of air. The residue was diluted in 1 mL dichloromethane and purified by solid-phase extraction. The determination of fatty acid methyl esters was performed with a GC instrument equipped with a flame ionization detector (Agilent 6850, Agilent Technologies, Waldbronn, Germany). Individual fatty acids were identified using the FAME C4-C24 mix (Supelco 18919) and additional individual fatty acid methyl esters. Quantification was performed via an internal standard. The HT content in the diets was determined as recently described by Johnson et al. [14].

### 2.5. Analysis of Boar Taint in Fat and Muscle

Androstenone, S, and I concentrations in the adipose tissue were analyzed according to Ampuero Kragten et al. [15]. The adipose tissue samples were liquefied in a microwave oven for 2 × 2 min at 250 W. The liquefied lipids were centrifuged for 2 min at room temperature. The water was then removed, and 0.5 mL of water-free liquid fat, kept at approximately 47 °C, was placed in 2.0 mL Eppendorf tubes in duplicates. For the quantification of extracted A, S, and I in the adipose tissue, an internal standard was added (1 mL methanol containing 0.469 mg/L androstanone and 0.05 mg/L 2-methylindole). After vortexing for 30 s, the tubes were incubated for 5 min at 30 °C in an ultrasonic-water bath, kept at 0 °C in an ice-water bath for 20 min and then centrifuged at 11,000 × *g* for 20 min. Finally, the liquid fraction was filtered (0.2 µm filter) and transferred into a vial for A, S, and I analysis with a high-performance liquid chromatography system. The concentrations were expressed per g of adipose tissue. The quantification limits were 0.3 µg/g tissue for A and 0.03 µg/g tissue for S and I.

### 2.6. RNA Isolation, Primer Design, and Quantitative Real-Time PCR

Total RNA extraction from the liver and colon mucosa was performed using Nucleospin^®^ RNA XS kit (740902, Macherey–Nagel, Oensinger, Switzerland) as previously described [5]. Concentrations were determined using a Nanodrop kit (Witec, Luzern, Switzerland) and quality assessed by capillary electrophoresis using a Fragment Analyzer (Advanced Instruments, Norwood, United States). To synthesize cDNA using a Verso cDNA synthesis kit (Thermo Scientific, Zug, Switzerland) 250 ng of RNA was used. The PCR was performed using a KAPA Sybr Fast qPCR Master Mix Universal on an Eco PCRmax real time PCR device (PCRmax, Staffordshire, United Kingdom). Primers for cytochrome (CYP)*1A1*, *CYP1A2*, *CYP2A19*, *CYP2E1*, and *CYP3A29* in both liver and colon mucosa were designed using the Primer-Blast service [16] offered by the National Institute of Health and verified for specificity using the National Center for Biotechnology Information database (www.ncbi.nlm.nih.gov). Primers for the *CYP1A2* gene were the same as those used by Rasmussen et al. [17]. The target and housekeeping gene primers and National Center for Biotechnology Information accession numbers are shown in Appendix A. For each primers pair, the efficiency of amplification was determined in three independent experiments. These genes were evaluated for their expression in pig livers and colons via quantitative real-time PCR, as outlined in detail by Bee et al. [5]. The amplification profile was composed of an activation step of 5 min at 95 °C, followed by 40 cycles of a two-step amplification (5 s at 95 °C and 20 s at 60 °C). The expression of each targeted gene was evaluated using the ΔΔ-Ct method (with efficiency corrections) and normalized using *GAPDH* as a housekeeping gene. All the calculations were performed using the Eco-Illumina Study software (Labgene, Chatel-St-Denis, Switzerland).

### 2.7. DNA Extraction and Sequencing

To extract bacterial DNA from the caecum samples, the QIAamp Fast DNA Stool Mini Kit (QIAGEN) was used, starting with 200 µg of samples following the manufacturers’ procedure. The DNA quality was assessed by capillary electrophoresis using a fragment analyzer (Advanced Analytics) and DNF-487 kit (Advanced Analytics). The extracted DNA was quantified using Nanodrop ND2000. The variable regions V3 and V4 of the 16S rRNA were amplified by PCR with universal primers for prokaryotic: (341F/802R: CCTACGGGNGGCWGCAG/GACTACHVGGGTATCTAATCC, respectively) (Takahashi et al. 2014). The PCR conditions were pre-denaturation at 95 °C for 3 min, 21 cycles of 95 °C for 30 s, 55 °C for 30 s, 72 °C for 30 s, and a final 5 min extension at 72 °C. The next-generation sequencing (NGS) of the extracted amplicons was performed by Microsynth AG (Balgach, Switzerland) on an Illumina MiSeq 300PE platform to obtain raw paired-end reads 2 × 300 bp.

### 2.8. NGS Data Analysis

Singleton reads were discarded and reference-based chimera were detected by UCHIME [18] based on the RDP classifier training database (v11) [19]. The operational taxonomic unit (OTU) table was achieved by mapping high-quality reads to the remaining OTUs with the Usearch [18] global alignment algorithm at a 97% cutoff. The 16S rRNA gene sequences, quality control and OTU binning were performed using the open source pipeline Quantitative Insights Into Microbial Ecology (QIIME) version 1.9.1 [20,21]. The sequences were binned into OTUs based on 97% identity against the Greengenes reference database (version 13.8) [22]. Sequence pre-processing, together with a microbial composition analysis and visualization at each taxonomic level, was defined using the “microbiome” package in R, version 3.5.0 (http://www.R-project.org). Alpha diversity was estimated using the Richness (Observed), Simpson and Shannon indices. Beta diversity was calculated using the weighted Unifrac distance method on the basis of the rarefied OTU abundance counts per sample. Additionally, the variance (PERMANOVA) and the similarities (ANOSIM) of the tested groups were analyzed. The alpha and beta diversity calculations and the rarefaction analysis were performed with the R software packages phyloseq v1.26.1 and vegan v2.5-5. To detect differentially abundant OTUs depending on collected sample metadata, differential OTU analysis on normalized abundance counts was performed with the R software package DESeq2 v1.22.2.

### 2.9. Statistical Analysis

Growth performance, carcass composition, meat quality traits and boar taint data were analyzed using the MIXED procedure of SAS (SAS Inst. Inc., Cary, NC, USA). The model included dietary PUFA content, HT level, and the PUFA × HT interaction as fixed effects, with litter as a random effect. Least squares means were calculated and considered statistically significant at *p* ≤ 0.05, and tendencies were denoted at *p* ≤ 0.10 and >0.05. Pearson correlations between the boar taint compounds and the weight of the testes, the accessory sex glands, and the expression of CYP gene isoforms were determined using the CORR procedure of SAS. A permutational multivariate analysis of variance (PERMANOVA) was used to evaluate whether the gut microbiota and the HT supplementation and/or the PUFA level differed (*p* < 0.05) among diets, also considering the effect size of the test (R). To identify bacterial taxa whose sequences were differentially abundant between the dietary treatments, emphasizing both statistical significance and biological relevance, a linear discriminant analysis of effect size was performed (LefSE) [23]. A LEfSe analysis was performed under the following conditions: the alpha value for the non-parametric factorial Kruskal–Wallis sum-rank test among the classes was <0.05, and the threshold on the logarithmic LDA score for the discriminative features was >4.0.

## 3. Results

### 3.1. Growth Performance, Feeding Behaviour, Carcass Characteristics, and Organ Weights

Supplementing the diet with HTs impaired (*p* < 0.01) the feed efficiency in both the grower and finisher periods. In the grower period, the impaired efficiency resulted from a similar growth rate but average daily feed intake tended to a greater (*p* = 0.10), whereas in the finisher period, the growth rate was impaired despite a similar feed intake (Table 2). The dietary PUFA level had no (*p* > 0.05) effects on the growth performance traits in the grower and finisher periods.

During the grower and finisher periods, the pigs fed the HT and soy oil-supplemented diet (H+) had fewer daily and total (*p* ≤ 0.09) feeder visits (Appendix A). However, these pigs stayed longer (*p* < 0.04) and ingested more feed per visit (only numerical differences in the H+ group, *p* = 0.11; soy oil group, *p* < 0.01) than their counterparts.

Except for the lighter (*p* = 0.001) livers, none of the organ weights, testes, carcass characteristics, or bulbo-urethral and salivary glands were affected by the HT intake or the dietary PUFA level (Appendix A). In accordance, no differences were observed in the bone, fat and lean mass of the half carcasses determined by the DXA (Appendix A).

### 3.2. Meat Quality, Boar Taint Compounds, and Cytochrome Isoenzyme Gene Expression

The loins of the pigs fed the HT supplemented diets were lighter (greater L* values; *p* = 0.05) and tended (*p* = 0.08) to be less red (lower a* values) compared to those fed the unsupplemented diets (Appendix A). Furthermore, the meat of pigs fed HTs (H+ and L+) had an overall lower (*p* = 0.04) water-holding capacity, which can be explained by the in tendency (*p* = 0.08) greater weight loss during cooking. These traits were unaffected by the dietary PUFA level. With respect to shear force, the pork of H+ pigs tended to be tougher compared to the L+ pigs, whereas intermediate values were observed in the pork of H− and L− pigs (T × P interaction; *p* = 0.07). In contrast, all the other traits were unaffected by the dietary treatments (Appendix A).

The levels of boar taint compounds were generally low and unaffected by the dietary PUFA level (Table 3). The I level tended (*p* = 0.08) to be lower in the EMs fed the HT-supplemented diet. The EMs that were fed the L− diet had lower S concentrations in the adipose tissue compared to those fed the H− diet, whereas intermediate levels were detected in the adipose tissue of H+ and L+ pigs (T × P interaction; *p* = 0.05).

The hepatic *CYP1A2* and *CYP2E1* isoenzyme expression levels in the EMs that were fed the HT-supplemented diets were lower (*p* = 0.04) and tended (*p* = 0.07) to be lower compared to those offered by the unsupplemented diets (Table 4). In the colon mucosa, the expression of *CYP3A29* tended (*p* = 0.06) to be lower when the EMs were fed the PUFA-enriched diets.

### 3.3. Fatty Acid Composition of the Intramuscular Fat and Adipose Tissue

Dietary HT supplementation had no impact on the fatty acid profile of the intramuscular fat and adipose tissue (Table 5). In contrast and as expected, the PUFA-enriched diets increased the degree of non-saturation of the intramuscular fat and, to a greater extent, that of the adipose tissue (Table 5). Common to both tissues, the 18:2n-6, 20:2n-6, 18:3n-3, and total PUFA levels were greater (*p* ≤ 0.02) and the 18:0 level was lower (*p* ≤ 0.04) in the H compared to the L groups. In addition, in the adipose tissue, the levels of 14:0, 16:0, 20:0, total SFA, 16:1n-7, 18:1n-9, 20:1n-7, and MUFA were lower (*p* ≤ 0.05) and the level of 20:3n-6, 22:5n-3, PUFA and the iodine value were greater (*p* < 0.01) in the EMs fed the H+ and H- diets compared to those fed the L+ and L- diets. Due to the larger difference in the amount of deposited n-3 (+189%) compared to n-6 (+137%) fatty acids, the n-6/n-3 fatty acid ratio was lower in the adipose tissue of pigs fed the H diets.

### 3.4. Effects of HTs and PUFA on Gut Microbiota

The genomic DNA purity and quality results are reported in Appendix A. A total of 5,172,998 paired-end 250-bp reads and 1786 OTUs were acquired. On average, 117,568.136 sequences per sample were obtained by 16S rRNA sequencing from the caecum content collected from each pig. An alpha diversity analysis showed that the Chao1 index, based upon the number of OTUs found in a sample, was lower (*p* = 0.001; Table 6), with average OTUs being consequently lower (*p* = 0.001) in H+ and L+ pigs compared to H− and L− pigs. Similarly, the phylogenetic diversity also decreased in the HT-supplemented groups. The dietary PUFA level did not affect microbiota alpha diversity, but the interaction between HTs and PUFA significantly (*p* = 0.03) affected the Shannon’s index.

The 16S rRNA datasets were then analyzed using UniFrac, an algorithm that measures the similarity between microbial communities based on the degree to which their component taxa share branch length on a bacterial tree of life [23]. For the unweighted UniFrac Beta diversity analysis, a clear (*p* = 0.01, R = 0.3) clusterization between diets based on the presence/absence of HTs was observed (Figure 1A). The amount of variation explained by the principal axes was 32.5% for PC1 and 10.7% for PC2. Similarly, the weighted UniFrac analysis revealed a less (*p* < 0.05) distinct clusterization of microbiota composition between the four dietary treatments based on the lower R value (0.19) (Figure 1B).

In this case, the amount of the full variation explained by the principal axes was greater than that observed in the unweighted analysis (44.2% and 17.1% for PC1 and PC2, respectively).

Because the dietary PUFA had no effects on the fecal microbiota composition, only the impact of HT levels on the differences in taxa abundance with biological relevance were considered for the linear discriminant analysis coupled with effect size measurements (LefSe). The analysis revealed that members of the Firmicutes phylum are affected. Dietary HT supplementation resulted in a greater proportion in the *Oscillospira* genus of the *Ruminococcaceae* family and lower amounts in the *Lactobacillales* order, *Streptococcaceae*, and *Veillonellaceae* families (Figure 2, Table 7).

Moreover, dietary HTs increased the number of *Treponema* and *Sphaerochaeta* genera, which are members of the *Spirochaetes* phylum, but reduced the *Proteobacteria* phylum abundance (Figure 2, Table 7).

## 4. Discussion

The primary goal of this study was to test the hypothesis proposed by Moerlein and Tholen [10] suggesting that the fatty acid composition of the adipose tissue can affect the level of A, S, and I, being lower in back fat rich in PUFA. We recently showed that bioactive compounds, such as HTs, have the potential to decrease the A, S, and I level [5]. However, the current findings do not suggest any relationship between boar taint level in the adipose tissue and the PUFA level in the back fat of EMs.

In addition, this study was meant to clarify which taxa are mainly affected by PUFA and HTs and/or its metabolites in order to estimate the potential associations between specific microbial taxa and A, S and I metabolism.

### 4.1. Dietary Effects on Growth Performance

In accordance with previous studies, the two iso-energetic and isonitrogenous grower and finisher diets differing in the fatty acid profile had no effect on the growth performance and the carcass characteristics [24,25]. In contrast, the dietary HT supply impaired feed efficiency in both the starter and finisher periods. In the grower period, this impairment was due to a similar growth rate and numerically greater feed intake, whereas the EMs fed the HT-supplemented diets grew slower in the finisher period despite a similar feed intake. A possible explanation for these findings is that HTs or their microbial degradation products negatively affected nutrient digestibility and absorption. Recent in vitro studies have revealed that HTs inhibit the pancreatic α-amylase activity and amylase absorption [26]. The current findings on feed efficiency are in agreement with those of previous studies [5,6] although they contradict the findings of Čandek–Potokar et al. [4]. The latter found a significant reduction in feed intake and a concomitant decrease in growth and thus similar feed efficiency with increasing HT levels. As in the present study, Bee et al. [5], and Čandek–Potokar et al. [4] used 3% chestnut HT extract in the diets. Therefore, the difference in feed efficiency between the studies might not be caused by the 3% HT extract, but by other nutritional factors or feed components. For instance, in the study of Čandek–Potokar et al. [4], the HT extract was added to a standard diet containing rapeseed meal. The latter if not being a “00” variety, could contain some antinutritional factors, such as erucic acid and glucosinolate. Furthermore, the coats of rapeseeds contain tannins and thus, with increasing HT extract supplementation, the total amount of tannins might have been sufficiently great to affect feed consumption.

### 4.2. Dietary Effects on Skatole, Indole, and Androstenone Levels in Adipose Tissue

Regardless of the feeding strategy, the A and S levels in the adipose tissue of the EMs were below the critical threshold values of 1 ppm for A and 0.25 ppm for S [1]. Nevertheless, two adipose tissue samples had A levels >1 ppm, one from the H− group and one from the H+ group. Furthermore, in three EMs, the concentration of S in the adipose tissue was >0.25 ppm, one from the H− and two from the L+ groups. None of the EMs had adipose tissues, which surpassed both the threshold values for A and S. A tendency for lower I and, although not significant, lower A levels was observed in the adipose tissues of the EMs fed the HT-supplemented diets, which concurs with previous findings [4,5]. A significant T × P interaction was found, which depicted lower S values in the adipose tissue of pigs fed the unsupplemented low-PUFA diet compared to those fed the unsupplemented high-PUFA diet. In contrast, when chestnut extract was offered, the S levels were intermediate regardless of the dietary PUFA supply. These findings are in contradiction with the results obtained from the analysis performed by Morlein et al. [10], as the S concentrations were lower in the L− compared to the H− group. A possible explanation for this contradiction is the fact that the overall average S concentrations were rather low in the present study.

### 4.3. Dietary Effects on the Fatty Acid Profile of the Adipose Tissue

The differences in the fatty acid profile of the adipose tissue were related to the dietary fat source, but not the chestnut extract supplementation. In contrast, Rezar et al. [27] observed greater PUFA levels in the subcutaneous fat of EM pigs fed a diet supplemented with 3% sweet chestnut wood extract. As explained by the authors, this effect was rather an indirect than direct effect of dietary sweet chestnut wood extract supplementation, as the supplement reduced the carcass fat deposition, resulting in a greater percentage of PUFA in the fat tissue. In the present study, the greater dietary PUFA intake decreased the relative concentration of the main SFA and MUFA in the adipose tissue of H+ and H− pigs. Although to a lesser extent, the effect of dietary PUFA intake was similar in the intramuscular fat. These effects were expected as it is well-known that increasing the dietary PUFA intake in pigs results in greater PUFA and concomitantly lower SFA and MUFA deposition in the fat tissues [28]. The latter is the result of downregulation of the fatty acid synthase and Δ-9-desaturase by elevated PUFA level in the tissues [24,29]. Increasing the PUFA tissue content has important implications on the technological and sensoric properties of the fat tissue, such as firmness oxidative stability. Wood et al. [28] reported that fat firmness is positively correlated with the level of 18:0 and negatively correlated with the level of 18:2n-6. In addition, with elevated tissue levels of unsaturated fatty acids, the susceptibility towards oxidation and thus rancidity increases [30]. To reduce the risk of oxidation, the study results suggested that PUFA levels higher than 18–50 g/kg and 22–23% in the feed and back fat, respectively, should be avoided [31,32]. To limit problems during processing due to soft adipose tissue, Swiss abattoirs control the PUFA level and iodine value of the adipose tissue at the slaughter line. When PUFA levels and iodine values exceed 15.5 g/100 fatty acids and 70, respectively, price markdowns per kg carcass are imposed [33]. Thus, Swiss feeding recommendations for pigs include two limits, one for PUFA and one for iodine value. Both use the weighed sum (expressed as g/kg) of SFA, MUFA, and PUFA. The limits are 5.1 and 7.8 g/kg feed for the PUFA and iodine values, respectively. In the present study, both the levels of PUFA in the H and L feed (0.6 and 0.4 g/kg, respectively) and in back fat (an average of 19.2% and 13.6% in H and L groups, respectively) were within the thresholds proposed by the Warnants et al. and Bryhni et al. [31,32]. In contrast, the pigs of the H treatment markedly surpassed the limits when the Swiss limits are applied.

### 4.4. Dietary Effects on the Faecal Microbial Composition

In the present study, dietary PUFA levels had no effects on the abundance and biodiversity of the microbial community, while the dietary HT level affected both traits. In fact, the EM pigs fed HTs showed a fecal microbiota with a lower abundance and alpha diversity indexes compared to the EM pigs fed without HTs. These results are in accordance with other authors who have described the effects of plant bioactive compounds on the gut bacteria growth and diversity [34,35,36]. A lower number of gut bacteria could affect the nutritional status and ultimately the weight gain of the growing pig given the reduced fermentation of undigested nutrients and consequently the lower production of SCFAs. Without this colonic microbiota activity, undigested nutrients would generally be eliminated via feces [37,38,39,40,41]. Likewise, Stanley et al. [42] observed similar evidence in chickens, where the feed-efficient birds showed greater microbial biodiversity as evidenced by markedly greater chao1 indexes than in less feed-efficient birds. Additionally, we observed in this study that HT supplementation not only affected the OTU number in the microbial ecosystem, but also changed the bacterial taxa composition. A beta diversity analysis determined by the weighted and unweighted UniFrac analysis showed a clear clusterization based on HT supplementation, but not PUFA levels. In this regard, dietary HTs increased the abundance of the *Oscillospira* genus. This genus has been linked to health traits both in humans and livestock and is known to produce butyric acid, which has been evaluated for the prevention of intestinal inflammation and colorectal cancer [43]. Furthermore, the abundance of *Treponema* and *Sphaerochaeta* genera was also greater in the HT-supplemented group. Conversely, HT supplementation decreased the abundance of the *Streptococcus* genus and *Proteobacteria*, their abundance being associated with an unhealthy gut [39]. In addition, HTs decreased the abundance of the *Veillonellaceae* family, which is known to be representative of both the human and pig microbiota, where some members of this family can be considered opportunistic pathogens [44]. Surprisingly, HTs reduced the abundance of the *Lactobacillales* order compared to the group with no dietary chestnut supply. Lactic acid bacteria are generally considered beneficial due to their exertion of protective functions, including antagonistic effects on gastroenteric pathogens, such as *Clostridium difficile, Campylobacter jejuni, Helicobacter pylori* and rotavirus [45]. Despite the decrease in the abundance of *Lactobacillales*, no visible impairment in the health status was detected independently of the dietary treatment.

### 4.5. Effects of HTs on the Gut Microbial Structure and Boar Taint

As demonstrated in this study, the addition of HTs inhibits gut microbial growth. Our hypothesis was that these changes in the microbial populations were related to a reduced microbial degradation of tryptophan to indole and skatole. A schematic model for a possible mode of action of dietary HTs on S and I synthesis and degradation is proposed in Figure 3.

It is known that gut cell debris is the major source of tryptophan for the microbial-mediated synthesis of indolic compounds [46,47]. In this regard, feed additives (e.g., antibiotics and Chinese herbs) which reduce the number of pathogenic bacteria in the intestine are effective in reducing the atrophy of villi in piglets after weaning [48]. As a consequence, the reduced abundance of the *Streptococcus* genus and *Proteobacteria*—associated with an unhealthy gut and inflammation [43], together with an increase of the *Oscillospira* genus and a negative correlation with intestinal inflammatory diseases [49,50]—should have led to a lower apoptosis of intestinal epithelial cells. Then, a lower availability of L-tryptophan from the cell debris and consequently microbial-mediated S and I production [6] were expected. Bacteria that degrade L-tryptophan are often capable of degrading other aromatic amino acids [51]. Several strains of *Lactobacillus* that produce S have been isolated from the bovine rumen and have been partially characterized; these organisms are able to produce S and I, not directly from L-tryptophan but by the decarboxylation of indole acetic acid [51]. Moreover, apart from S, these Lactobacillus strains are also produced by the same mechanism of decarboxylation p-cresol (4-methylphenol), another major phenolic component of pig odour [51]. Despite the changes in the gut microbiota composition due to the presence of HTs, little effects have been observed on the boar taint level in the adipose tissue. As gut inflammation markers and the gut cell mitotic rate have not been considered in this study, it is not possible to investigate the differences in the intestinal cell debris due to the HT supplementation. One can hypothesize that the chemical composition of the tested diets had a similar effect on gut cell mitosis and ultimately on S and I formation from the tryptophan in the pig colon.

## 5. Conclusions

These data do not confirm that the boar taint compound levels were related to the PUFA levels in the adipose tissue. However, supplementing the diet of the EMs with 3% chestnut wood extract containing HTs affected both, qualitatively and quantitatively, the pig gut microbiota without having an impact on its biodiversity, independent of the dietary PUFA levels. It is still unclear whether HTs themselves or their derivatives exert the observed effects on gut microbiota. Consequently, further investigations are needed to clarify the nature of the association between HTs, their derivatives, the gut microbial ecosystem and the synthesis of S and I.

## Figures and Tables

**Figure 1 animals-09-01063-f001:**
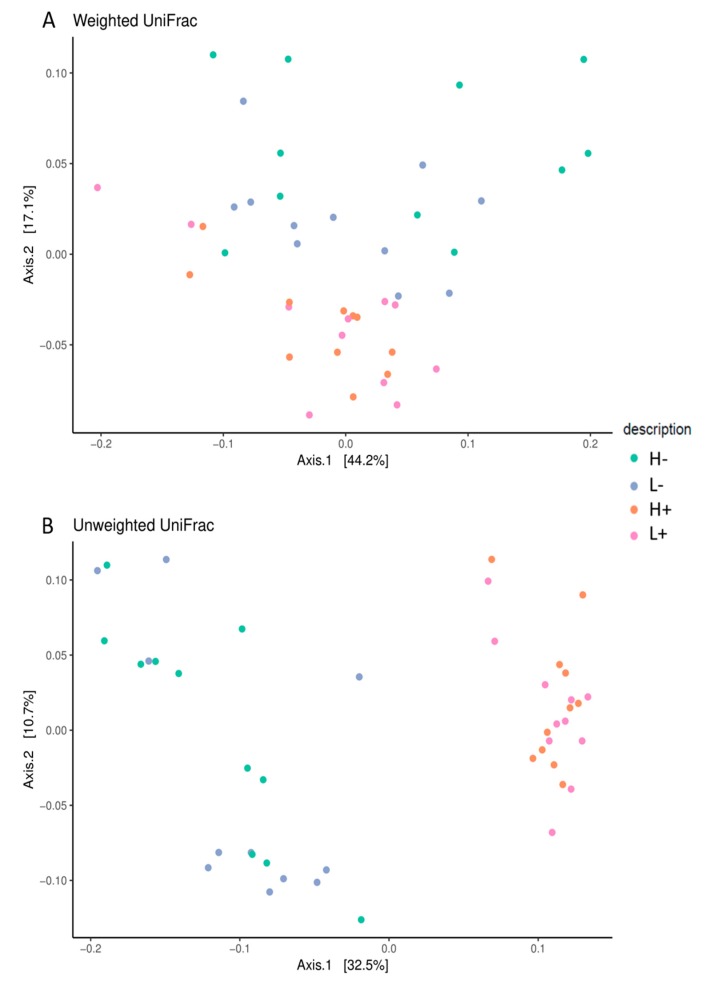
Beta diversity analysis: Unweighted (**A**), and weighted (**B**) UniFrac principal component analysis (PCA) of caecum microbiota collected from pigs fed PUFA-enriched diet supplemented with 3% chestnut extract (H+; N = 12), PUFA-enriched diet supplemented with 0% chestnut extract (H−; N = 12), PUFA-reduced diet with 3% chestnut extract (L+, N = 12), and PUFA-reduced diet with 0% chestnut extract (L−, N = 12).

**Figure 2 animals-09-01063-f002:**
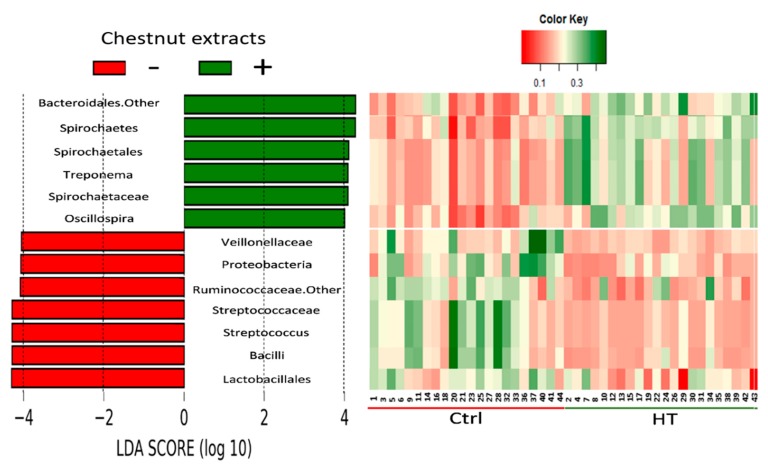
Linear discriminant analysis coupled with effect size measurements (LefSe): The most differentially abundant genus level taxa determined in caecum samples from pigs fed diets without chestnut extract supplementation (−; in red, N = 22) or the diets supplemented with 3% chestnut extract containing hydrolysable tannins (+; in green, N = 22). The heat map shows the scores of these relative abundance levels.

**Figure 3 animals-09-01063-f003:**
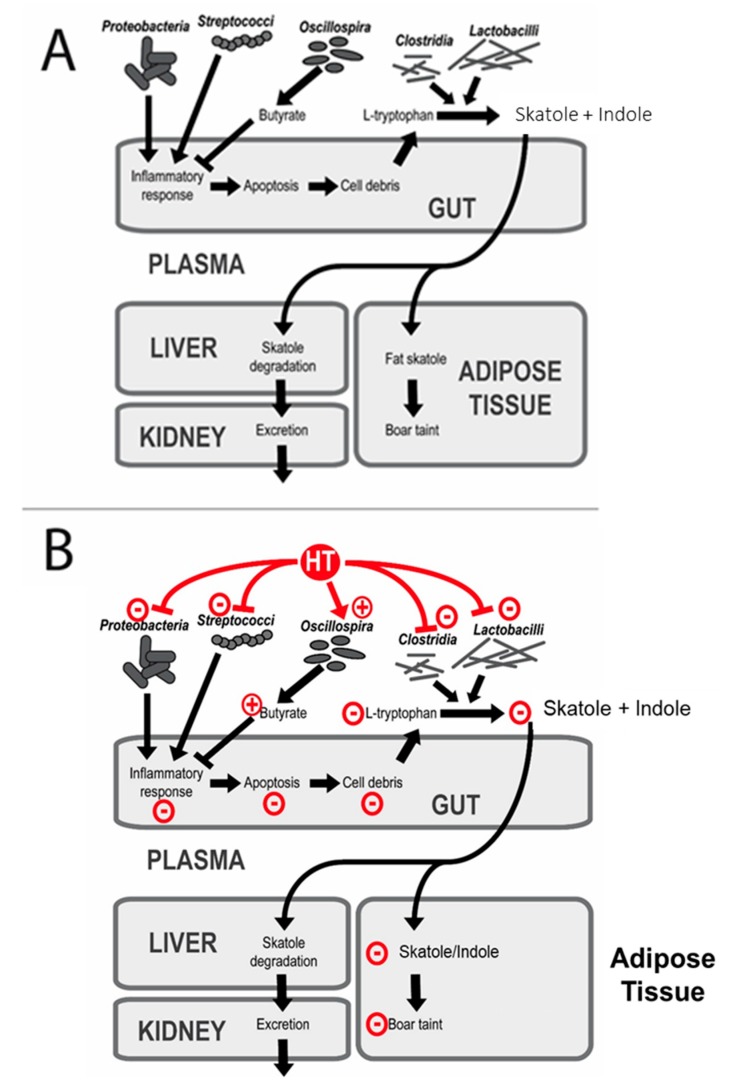
Proposed model for the actions of dietary hydrolysable tannins (HTs) on skatole synthesis and degradation. Panel (**A**): Skatole is produced via the fermentation of L-tryptophan (Trp) by the gut microbiota [46]. The last step of the pathway is performed by the decarboxylases of several *Clostridiales* (*Clostridium scatologenes*, *Clostridium nauseum*) and several *Lactobacillales* (*Lactobillus helveticus, Lactobillus sp* 11201), among others [51]. Tryptophan originates from the naturally occurring cell debris of apoptotic cells, and its amount is increased by the inflammatory response in the gut [6]. Skatole is then transferred from the gut to the plasma where it can be either incorporated into adipose tissues or degraded in the liver and subsequently excreted by the kidney. Panel (**B**): Dietary HTs and/or their metabolites positively enhance the proportion of the *Oscillospira* species in the gut. *Oscillospira*-mediated butyrate inhibits the inflammatory response in the gut, thus leading to less cell debris and tryptophan available for S production. In addition, dietary HTs reduce the proportion of *Streptococci* and *Proteobacteria* in the gut, thus limiting the *Streptococci* and *Proteobacteria*-born inflammatory response. Moreover, dietary HTs decrease the amounts of *Lactobacillaceae*, which are involved in S production from tryptophan.

**Table 1 animals-09-01063-t001:** Composition of the experimental diets, as-fed basis ^1^.

Item	Grower Diet (from 73 to 122 d of Age)				Finisher Diet (from 122 to 170 d of Age)			
	H−	L−	H+	L+	H−	L−	H+	L+
Wheat	49.75	49.75	49.75	49.75	42.59	42.59	42.59	42.59
Barley	10.56	10.56	10.56	10.56	27.7	27.7	27.7	27.7
Corn	2.43	2.43	2.43	2.43				
Wheat starch	7	7	7	7	9.09	9.09	9.09	9.09
Soy extraction meal	12.71	12.71	12.71	12.71	9.7	9.7	9.7	9.7
Potato protein	2.47	2.47	2.47	2.47	0.59	0.59	0.59	0.59
Wheat bran	5	5	5	5	1.14	1.14	1.14	1.14
Soy oil	2	-	2	-	2	-	2	-
Tallow	-	2	-	2	-	2	-	2
Arbocel	3	3	-	-	3	3	-	-
Hydrolysable tannins ^2^	-	-	3	3	-	-	3	3
Dicalcium phosphate	1.45	1.45	1.45	1.45	1.05	1.05	1.05	1.05
Mono-sodium phosphate	0.4	0.4	0.4	0.4	-	-	-	-
Calcium carbonate	1.38	1.38	1.38	1.38	0.88	0.88	0.88	0.88
NaCl	0.08	0.08	0.08	0.08	0.46	0.46	0.46	0.46
Natuphos 5000 G	0.01	0.01	0.01	0.01	0.01	0.01	0.01	0.01
L-lysine-HCl	0.34	0.34	0.34	0.34	0.36	0.36	0.36	0.36
DL-methionine	0.04	0.04	0.04	0.04	0.04	0.04	0.04	0.04
L-threonine	0.07	0.07	0.07	0.07	0.1	0.1	0.1	0.1
Mikrogrit	0.6	0.6	0.6	0.6	0.6	0.6	0.6	0.6
Pellan ^3^	0.3	0.3	0.3	0.3	0.3	0.3	0.3	0.3
Mineral-vitamin premix ^4^	0.4	0.4	0.4	0.4	0.4	0.4	0.4	0.4
Analyzed nutrient and tannin composition, g/kg DM								
Total ash	59	59.9	59.6	59.3	48.6	48.1	49.3	49
Crude fiber	46.2	44.7	27.7	27	45.1	44.9	32.6	31.5
Crude protein	177.1	180.7	180.1	180.5	146.4	146.9	147.3	145.4
Crude fat	42.3	44.5	42.3	45.6	41.3	42.9	42.3	45
SFA (g/100 total fatty acid)	38.46	42.91	38.46	42.91	37.31	42.45	37.31	42.45
MUFA (g/100 total fatty acid)	32.69	44.98	32.69	44.98	32.34	45.32	32.34	45.32
PUFA (g/100 total fatty acid)	28.85	12.11	28.85	12.11	30.35	12.23	30.35	12.23
Total hydrolysable tannin	-	-	14.81	14.81	-	-	14.81	14.81
Calculated DE content, MJ/kg DM ^5^	13.54	13.54	13.54	13.54	13.54	13.54	13.54	13.54

^1^ Grower diet formulated for pigs in the BW range of 25 to 60 kg; finisher diet formulated for pigs in the BW range of 60 to 110 kg; H = diets that had a high PUFA content and were supplemented without (H−), or with (H+) a chestnut powder containing hydrolysable tannin. L = diets that had a low PUFA content and were supplemented without (H−) or with (H+) a chestnut powder containing hydrolysable tannin. ^2^ Chestnut extract, containing. ^3^ Binder that aids in pellet formation. ^4^ Supplied the following nutrients per kg of diet: 20000 IU vitamin A, 200 IU vitamin D3, 39 IU vitamin E, 2.9 mg riboflavin, 2.4 mg vitamin B6, 0.010 mg vitamin B12, 0.2 mg vitamin K3, 10 mg pantothenic acid, 1.4 mg niacin, 0.48 mg folic acid, 199 g choline, 0.052 mg biotin, 52 mg Fe as FeSO_4_, 0.16 mg I as Ca(IO)3, 0.15 mg Se as Na_2_Se, 5.5 mg Cu as CuSO_4_, 81 mg Zn as ZnO_2_, and 15 mg Mn as MnO_2_. ^5^ The digestible energy coefficients from each feed ingredient were obtained from the Swiss Feed Database (https://www.feedbase.ch), and taking into account the relative amount of each feed ingredient in the diet, the digestible energy content was calculated. Abbreviations: PUFA, polyunsaturated fatty acid; MUFA, monounsaturated fatty acid; SFA, saturated fatty acids; BW, body weight; DE, digestible energy; DM, dry matter.

**Table 2 animals-09-01063-t002:** Effect of dietary hydrolysable tannin and PUFA level on growth performance of grower–finisher pigs ^1^.

Item	Dietary Treatments				SEM	*p*-Values ^2^		
	H−	L−	H+	L+		T	*P*	T × P
Body weight, kg								
At birth	1.68	1.64	1.75	1.71	0.109	0.36	0.64	0.98
At start of grower period	25.60	25.42	26.51	26.64	1.553	0.42	0.98	0.91
At start of finisher period	66.45	62.60	65.86	66.83	2.377	0.44	0.54	0.31
At slaughter	112.76	109.00	108.49	110.40	3.791	0.63	0.76	0.34
Average daily gain, kg/d								
Grower period	0.83	0.76	0.80	0.82	0.029	0.58	0.46	0.14
Finisher period	0.94	0.94	0.87	0.88	0.037	0.01	0.94	0.71
Grower–finisher period	0.89	0.85	0.83	0.85	0.029	0.27	0.69	0.27
Total feed intake, kg								
Grower period	83.8	77.9	84.6	86.5	2.831	0.10	0.48	0.17
Finisher period	124.1	119.4	118.2	118.4	6.571	0.34	0.53	0.49
Grower–finisher period	207.7	197.5	202.9	205.3	8.535	0.80	0.50	0.28
Average feed intake, kg/d								
Grower period	1.71	1.59	1.73	1.77	0.058	0.10	0.48	0.17
Finisher period	2.51	2.43	2.40	2.40	0.093	0.33	0.56	0.56
Grower–finisher period	2.11	2.01	2.06	2.09	0.071	0.82	0.51	0.31
Gain-to-feed, kg/kg								
Grower period	0.49	0.48	0.47	0.47	0.008	0.01	0.34	0.34
Finisher period	0.38	0.39	0.36	0.37	0.008	<0.01	0.19	0.63
Grower–finisher period	0.42	0.42	0.41	0.41	0.006	<0.01	0.56	0.78

^1^ H−: high amount of polyunsaturated fatty acid (PUFA) without chestnut extract containing HTs; H+: high PUFA with 3% chestnut extract containing HTs; L−: low PUFA without chestnut extract containing HTs; L+: low PUFA with 3% chestnut extract containing HTs (L+). ^2^ Probability values for hydrolysable tannin supplementation (T), dietary PUFA level (P), and T × P interaction.

**Table 3 animals-09-01063-t003:** Effect of dietary hydrolysable tannin and PUFA level on androstenone, skatole, and indole levels in the adipose tissue of grower–finisher pigs ^1^.

Item	Dietary Treatments				SEM	*p*-Values ^2^		
	H−	L−	H+	L+		T	*P*	T × P
Boar taint compounds, µg/g adipose tissue								
Androstenone	0.51	0.41	0.37	0.39	0.085	0.32	0.62	0.40
Skatole	0.13 ^y^	0.05 ^x^	0.09 ^x,y^	0.12 ^x,y^	0.026	0.48	0.25	<0.05
Indole	0.05	0.03	0.02	0.02	0.007	0.08	0.16	0.21

^x,y^ Values within a row with different superscripts tend to differ significantly at *p* ≤ 0.10. ^1^ H−: high amount of polyunsaturated fatty acid (PUFA) without chestnut extract containing HTs; H+: high PUFA with 3% chestnut extract containing HTs; L−: low PUFA without chestnut extract containing HTs; L+: low PUFA with 3% chestnut extract containing HTs (L+). ^2^ Probability values for hydrolysable tannin supplementation (T), dietary PUFA level (P), and T × P interaction.

**Table 4 animals-09-01063-t004:** Effect of hydrolysable tannin and PUFA in the growing and finisher diets of entire males on hepatic mRNA cytochrome P450 isoenzyme expression ^1^.

Item	Dietary Treatments				SEM	*p*-Values ^2^		
	H−	L−	H+	L+		T	*P*	T × P
	LIVER							
*CYP1A1* *	1.11	1.19	0.93	1.22	0.285	0.83	0.69	0.53
*CYP1A2*	1.59	1.24	0.93	1.03	0.200	0.04	0.52	0.28
*CYP2A19* *	2.08	1.62	2.26	1.08	0.812	0.21	0.13	0.23
*CYP2E1*	1.15 ^x,y^	1.45 ^y^	1.13 ^x,y^	1.12 ^x^	0.166	0.07	0.22	0.11
*CYP3A29*	1.43	1.24	1.25	1.02	0.109	0.40	0.36	1.00
	Colon							
*CYP1A1*	0.10	0.79	1.25	0.18	0.566	0.59	0.71	0.09
*CYP3A29* *	1.14	1.55	0.10	1.17	0.075	0.14	0.06	0.42

x,y Values within a row with different superscripts tend to differ significantly at *p* ≤ 0.10. * As data were not normally distributed, they were log transformed prior to analysis. Results presented are back-transformed data. ^1^ H−: high amount of polyunsaturated fatty acid (PUFA) without chestnut extract containing HTs; H+: high PUFA with 3% chestnut extract containing HTs; L−: low PUFA without chestnut extract containing HTs; L+: low PUFA with 3% chestnut extract containing HTs (L+). ^2^ Probability values for hydrolysable tannin supplementation (T), dietary PUFA level (P), and T × P interaction.

**Table 5 animals-09-01063-t005:** Effect of hydrolysable tannin and PUFA in the growing and finisher diet of entire males on intramuscular fat and adipose tissue fatty acid composition ^1^.

Item	Dietary Treatments				SEM	*P*-Values ^2^		
	H−	L−	H+	L+		T	*P*	T × P
Intramuscular fat (g/kg)	22.30	19.02	21.12	20.59	1.882	0.88	0.25	0.43
Fatty acid profile (g/100 g total fatty acid)								
14:0	1.03	0.91	0.98	0.99	0.074	0.81	0.22	0.13
16:0	22.9	22.7	22.8	23.0	0.389	0.79	0.94	0.24
17:0	0.20	0.20	0.20	0.20	0.031	0.92	0.98	0.80
18:0	11.8 ^x^	12.4 ^y^	12.2 ^x,y^	12.5 ^y^	0.288	0.31	0.04	0.46
20:0	0.07	0.06	0.07	0.07	0.025	0.82	0.86	0.68
16:1n-7	3.29 ^y^	3.27 ^y^	3.02 ^x^	3.30 ^y^	0.125	0.12	<0.01	0.05
17:1n-7	0.20	0.20	0.18	0.22	0.030	0.94	0.33	0.34
18:1n-9	45.3	45.9	45.0	46.3	0.539	0.96	0.08	0.47
20:1n-9	0.67	0.58	0.65	0.68	0.040	0.27	0.46	0.12
18:2n-6	10.9	10.3	11.4	9.65	0.661	0.80	0.02	0.26
18:3n-3	0.74	0.40	0.70	0.43	0.041	0.89	<0.01	0.19
20:2n-6	0.28	0.17	0.25	0.21	0.052	0.83	0.01	0.15
20:4n-6	2.04	2.52	2.16	2.11	0.203	0.40	0.23	0.14
22:4n-6	0.18	0.17	0.13	0.17	0.058	0.51	0.56	0.54
22:5n-3	0.21	0.14	0.18	0.17	0.059	0.95	0.31	0.54
16:1n-7/16:0	0.14 ^y^	0.14 ^y^	0.13 ^x^	0.14 ^y^	0.005	0.08	0.07	0.09
18:1n-9/18:0	3.86	3.72	3.71	3.74	0.094	0.40	0.43	0.29
20:4n-6/18:2n-6	0.19	0.24	0.19	0.22	0.012	0.27	<0.01	0.22
22:5n-3/18:3n-3	0.29	0.32	0.25	0.35	0.101	0.96	0.35	0.63
Sum of n-6 fatty acids	13.45	13.21	13.94	13.15	0.860	0.66	0.13	0.24
Sum of n-3 fatty acids	0.96	0.54	0.88	0.60	0.083	0.76	<0.01	0.13
SFA	36.1	36.3	36.2	36.7	0.592	0.48	0.33	0.75
MUFA	49.5	49.9	48.9	50.5	0.586	0.93	0.07	0.29
PUFA	14.4	13.8	14.8	12.7	0.899	0.66	0.05	0.29
n-6/n-3 fatty acid ratio	14.6 ^x^	24.0^z^	17.8 ^y^	23.1 ^z^	2.022	0.31	<0.01	0.07
18:2n-6/18:3n-3 ratio	14.9 ^x^	23.2 ^z^	17.1 ^y^	22.8 ^z^	1.125	0.18	<0.01	0.07
Adipose tissue								
Fatty acid profile (g/100 g total fatty acid)								
14:0	1.19	1.30	1.16	1.31	0.025	0.61	<0.01	0.43
16:0	22.5	24.0	22.2	24.0	0.294	0.57	<0.01	0.58
18:0	12.2	13.2	11.9	13.1	0.413	0.54	<0.01	0.75
20:0	0.19	0.17	0.17	10.1	0.006	0.10	0.01	0.91
16:1n-7	1.98	2.43	1.90	2.48	0.074	0.75	<0.01	0.21
18:1n-9	41.3	43.6	41.4	43.1	0.404	0.51	<0.01	0.44
20:1n-9	0.85	0.95	0.83	0.88	0.039	0.22	0.05	0.42
18:2n-6	15.9	11.5	16.5	11.9	0.568	0.24	<0.01	0.92
20:2n-6	0.64	0.50	0.66	0.49	0.025	0.61	<0.01	0.39
18:3n-3	1.80	0.86	1.87	0.94	0.053	0.15	<0.01	0.90
20:3n-3	0.25	0.15	0.26	0.16	0.010	0.37	<0.01	0.95
20:4n-6	0.24	0.22	0.22	0.21	0.011	0.21	0.08	0.98
22:4n-3	0.07	0.07	0.06	0.06	0.005	0.08	0.82	0.34
22:5n-3	0.10	0.07	0.10	0.07	0.006	0.88	<0.01	0.93
16:1n-7/16:0	0.09	0.10	0.09	0.10	0.003	0.91	<0.01	0.42
18:1n-9/18:0	3.44	3.33	3.54	3.31	0.127	0.71	0.14	0.60
20:4n-6/18:2n-6	0.01	0.02	0.01	0.02	0.001	<0.01	<0.01	0.82
22:5n-3/18:3n-3	0.06	0.09	0.05	0.08	0.004	0.24	<0.01	0.491
SFA	36.5	39.2	35.9	39.2	0.647	0.55	<0.01	0.62
MUFA	44.5	47.4	44.4	46.9	0.428	0.46	<0.01	0.54
PUFA	18.9	13.3	19.6	13.9	0.661	0.26	<0.01	0.92
n-6/n-3 fatty acid ratio	7.81 ^x^	11.4 ^z^	7.81 ^x^	10.9 ^y^	0.132	0.07	<0.01	0.06
18:2n-6/18:3n-3 ratio	8.81 ^x^	13.4 ^z^	8.81 ^x^	12.8 ^y^	0.174	0.05	<0.01	0.05
Iodine value	72.1	64.4	73.2	64.9	1.100	0.37	<0.01	0.77

^x,y,z^ Values within a row with different superscripts tend to differ significantly at *p* ≤ 0.10. ^1^ H = high dietary PUFA level by including 2% soy oil; L = low dietary PUFA supplementation by including 2% tallow; − = without hydrolysable tannin supplementation; + = with hydrolysable tannin (3%) supplementation. ^2^ Probability values for hydrolysable tannin supplementation (T), dietary PUFA level (P), and T × P interaction.

**Table 6 animals-09-01063-t006:** Summary of next-generation sequencing data and effect of different dietary treatments on diversity and abundance indexes in entire male pigs ^1^.

Item	Dietary Treatments				*p*-Values ^2^		
	H−	L−	H+	L+	T	*P*	T × P
Chao1 ^3^	1002 ± 94.7	1017 ± 120	900 ± 83.7	912 ± 63.9	<0.01	0.64	0.96
OTUs ^4^	916 ± 93.8	944 ± 118	831 ± 74.8	840 ± 54.8	<0.01	0.47	0.75
Shannon ^5^	4.89 ± 0.41 ^x^	5.17 ± 0.23 ^y^	5.07 ± 0.12 ^x,y^	5.02 ± 0.11 ^x,y^	0.86	0.15	0.03
PD ^6^	55.5 ± 3.21	57.1 ± 4.66	52.3 ± 3.27	52.8 ± 2.31	<0.01	0.31	0.63

^x,y^ Values within a row with different superscripts differ significantly. ^1^ H−: high amount of polyunsaturated fatty acid (PUFA) with 0% chestnut extract supplementation; H+: high PUFA with 3% chestnut extract containing HTs; L−: low PUFA without chestnut extract containing HTs; L+: low PUFA with 3% chestnut extract containing HTs (L+). ^2^ Probability values for hydrolysable tannin supplementation (T), dietary PUFA level (P), and T × P interaction. ^3^ Chao1: bacterial community index. ^4^ OTUs: operational taxonomic units. ^5^ Shannon: Shannon diversity index. ^6^ PD: phylogenetic diversity index.

**Table 7 animals-09-01063-t007:** Relative abundance levels (%) of the most significant different phylotypes between pigs fed diets without tannins (−, N = 22) and diets supplemented with 3% chestnut extract containing hydrolysable tannins (+, N = 22) ^1^.

Taxonomic Rank		HT		*p*-Value
		−	+	
Phylum	*Bacteroidetes*			
Family	*Unclassified Bacteroidales*	0.106	0.182	<0.001
Phylum	*Spirochaetes*	0.058	0.096	<0.001
Order	*Spirochaetales*	0.026	0.051	<0.001
Family	*Spirochaetaceae*	0.025	0.050	<0.001
Genus	*Treponema*	0.025	0.049	<0.001
Phylum	*Firmicutes*			
Class	*Bacilli*	0.053	0.014	<0.001
Family	*Veillonellaceae*	0.047	0.025	<0.001
Genus	*Unclassified Ruminococcaceae*	0.077	0.052	0.002
Genus	*Oscillospira*	0.022	0.042	<0.001
Order	*Lactobacillales*	0.051	0.011	<0.001
Family	*Streptococcaceae*	0.045	0.007	<0.001
Genus	*Streptococcus*	0.045	0.007	<0.001
Phylum	*Proteobacteria*	0.051	0.028	<0.001

^1^ Data are expressed as the percentage of the relative abundance of all sequences in each group.

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
