# Peer review of "Effect of Supplementing Hydrolysable Tannins to a Grower–Finisher Diet Containing Divergent PUFA Levels on Growth Performance, Boar Taint Levels in Back Fat and Intestinal Microbiota of Entire Males"

_animals, 2019, doi:10.3390/ani9121063_

Round 1

Reviewer 1 Report

The paper is an interesting exercise on the effect of supplementst o the diet.

It is not clear however how AAs proposed the model of Figure 3.

Moreover, the comments on gut microbiota relay basically on a software but a more intelligible explanation would be welcomed.

Reviewer 2 Report

The manuscript describe the effect of a combination of PUFA and hydrolysable Tannins  in the growing-finishing diet for pigs, with the objective to verify a synergistic effect in reducing the boar taint in entire male.

The trial is properly designed to answer the posed questions. Maybe some tables included as supplementary (like feeding behavior) could be included in the main text and discussed. Finally, the manuscript is well written. Some minor  implementation are requested as listed below:   

Line 103-104: did you use 4 pigs/litter? Did balanced the experimental groups on the bases of the litter? Please, specify the weaning age of the pigs as well as the pig’s age at the beginning of the trial.

Line 108: can you specify the origin of the PUFA included in the diet, please? 

Line 128: Here is not clear how long pigs received the diets. What you mean with “for 92 to 104 d?

Line 134: 44 pigs?

Line 198: please better describe the protocol for the qPCR, at least the used polymerase and the instrument.

Line 232: please specify the version of Greengenes used.

Line 256: please, modify grater, is just a tendency.

Line 267: please use the same acronyms along the text and in the tables (e.g.  HT- is  not in line with the acronyms reported in line 105-108 ad in the Tables).

Line 225: The bioinformatics procedures for microbiota analyses should be deeply described (e.g. clarify how you define the alpha and beta diversity, Taxonomic differences, etc). Please include all the packages used.

Line 279: P=0.08 can not be associate with “greater”

Line 309: non-saturation – better unsaturated

Line 325: are you sure all the DNA are bacterial DNA? Is not possible to discriminate the genomic DNA deriving from the host and the bacterial DNA just with the Nanodrop (this could be possible using the digital PCR). Did you performed a DNA treatment to remove the host DNA?  

The quality of the figure 1 and 2 should be improved.

Line 388-401: this part is redundant. The objective has been already listed previously. Here the authors can summarize in few lines the idea at the bases of the project.

Line 402: please, delete “dietary”

Line 421-30: the authors asserted the reduction in microbial abundance  and diversity in HT groups can  explain the impairment of the feed utilization in these groups. From another hand as observe with  the AGPs, the reduction of the gut microbiota, could be lied with the increasing of the growth performance, due to the reduction of the  competition between the host and the microbiota for the nutrients as well as a lower activation of the immune system. Do you consider the effect of HT on the feed digestibility due to “antinutritional” effect of this compounds, especially when added at high dose (3%)?

 Can the authors reported data on  the percentage of the body fat of pigs feed with HT, please?

Can the authors explain the choice to analyze the cecum microbiota instead of Jejunum microbiota? Do you expected an action  of the Tannins in the large intestine?

The authors collected also ascending and descending colon content, but in the manuscript are not reported any data related with this samples, is there a reason for that?  

Table 1: please include the range of days for each time period (e.g from dx to day of age)

Supplementary Table 1: please include the fragment length and the annealing temperature for each gene.

In the “discussion” the authors hyotized a linkage between the modification of the cecum microbial profile in the HT groups and the observed reduction of the growth performance in these groups. While in the “conclusion” they asserted: “This study confirms that HTs have antibacterial activity with no negative effects on the bacterial community” . A microbial profile that do not allow to maximize the diet utilization impairing  the growth performance, could be defined “beneficial” or "optimal" in a due situation. Can  you better support your conclusions. 

Reviewer 3 Report

This is an interesting study and the authors have collected a very good dataset using good methodology. The paper is generally well structured and scientifically sound.
